# Unified inference of missense variant effects and gene constraints in the human genome

Yi-Fei Huang[1,2]*

**1** Department of Biology, Pennsylvania State University, University Park, Pennsylvania, United States of America, **2** Huck Institutes of the Life Sciences, Pennsylvania State University, University Park, Pennsylvania, United States of America

* yuh371@psu.edu

**Data Availability Statement:** The UNEECON program and precomputed UNEECON/UNEECON-G scores are available at https://github.com/yifei-lab/UNEECON.

**Funding:** This work was supported by start-up funds from the Pennsylvania State University. The

## Abstract

A challenge in medical genomics is to identify variants and genes associated with severe genetic disorders. Based on the premise that severe, early-onset disorders often result in a reduction of evolutionary fitness, several statistical methods have been developed to predict pathogenic variants or constrained genes based on the signatures of negative selection in human populations. However, we currently lack a statistical framework to jointly predict deleterious variants and constrained genes from both variant-level features and gene-level selective constraints. Here we present such a unified approach, UNEECON, based on deep learning and population genetics. UNEECON treats the contributions of variant-level features and gene-level constraints as a variant-level fixed effect and a gene-level random effect, respectively. The sum of the fixed and random effects is then combined with an evolutionary model to infer the strength of negative selection at both variant and gene levels. Compared with previously published methods, UNEECON shows improved performance in predicting missense variants and protein-coding genes associated with autosomal dominant disorders, and feature importance analysis suggests that both gene-level selective constraints and variant-level predictors are important for accurate variant prioritization. Furthermore, based on UNEECON, we observe a low correlation between gene-level intolerance to missense mutations and that to loss-of-function mutations, which can be partially explained by the prevalence of disordered protein regions that are highly tolerant to missense mutations. Finally, we show that genes intolerant to both missense and loss-of-function mutations play key roles in the central nervous system and the autism spectrum disorders. Overall, UNEECON is a promising framework for both variant and gene prioritization.

## Author summary

Numerous statistical methods have been developed to predict deleterious missense variants or constrained genes in the human genome, but unified prioritization methods that utilize both variant- and gene-level information are underdeveloped. Here we present UNEECON, an evolution-based deep learning framework for unified variant and gene prioritization. By integrating variant-level predictors and gene-level selective constraints,

funder had no role in study design, data collection and analysis, decision to publish, or preparation of the manuscript.

**Competing interests:** The author has declared that no competing interests exist.

UNEECON outperforms existing methods in predicting missense variants and protein-coding genes associated with dominant disorders. Based on UNEECON, we show that disordered proteins are tolerant to missense mutations but not to loss-of-function mutations. In addition, we find that genes under strong selective constraints at both missense and loss-of-function levels are strongly associated with the central nervous system and the autism spectrum disorders, highlighting the need to investigate the function of these highly constrained genes in future studies.

## Introduction

A fundamental question in biology is to understand how genomic variation contributes to phenotypic variation and disease risk. While millions of protein-altering variants have been identified in the human genome, it is challenging to assess the functional and clinical significance of these variants. In particular, a large fraction of missense variants have been annotated as "variants of uncertain significance" (VUS) [1, 2], forming a major hurdle for both basic research and medical practice. This problem is further exacerbated by the difficulty of experimentally validating the function of large numbers of missense variants *in vivo*. Therefore, there is a tremendous need for accurate computational tools to prioritize deleterious missense variants [3].

Since early-onset, severe genetic disorders are often associated with a reduction of evolutionary fitness, signatures of negative (purifying) selection, such as sequence conservation, have been widely used to predict deleterious variants associated with Mendelian disorders [4–12]. Among existing evolutionary approaches for variant prioritization, recently developed integrative methods are particularly powerful [8–15]. By learning a linear or nonlinear mathematical function from predictive variant features, such as sequence conservation scores and protein structural features, to the strength of negative selection, these statistical methods estimate negative selection on observed and potential mutations in the human genome. The estimated strength of negative selection can then be utilized to prioritize deleterious variants associated with severe genetic disorders. Because these evolutionary approaches are trained on tremendous natural polymorphisms observed in healthy individuals instead of sparsely annotated pathogenic variants, they have shown good performance in predicting pathogenic variants, frequently outperforming or on par with supervised machine learning models trained on disease data [8, 10–12].

Despite the success of evolution-based metrics of variant effects, the existing methods nevertheless suffer from a few critical limitations. First, most existing methods focus on learning a shared mathematical function from predictive variant features to negative selection and assume that this function is equally applicable to all protein-coding genes. Instead, a subset of genes can depart from the genome-wide trend between variant features and negative selection, possibly due to enhanced or relaxed purifying selection on these genes in the human lineage [12]. Second, most existing methods are trained on common genetic variants, making it challenging to distinguish strong negative selection associated with severe genetic diseases from moderate negative selection without clinical implications. Third, these methods typically are agnostic to the mode of inheritance and, therefore, may be suboptimal in the prediction of deleterious variants associated with dominant disorders on which we focus in this work.

In parallel with the development of variant-level interpretation methods, several complementary, gene-centric methods have been proposed to predict protein-coding genes associated with dominant genetic disorders [16–23]. Unlike variant-level predictors, the gene-level

prioritization methods seek to identify constrained genes that are intolerant to heterozygous nonsynonymous mutations. These gene-level constraint metrics have been shown to provide complementary information on variant effects and have been successfully used to prioritize variants associated with dominant genetic disorders [16, 17, 24]. However, these methods typically assume that all the missense mutations in a gene or a genic region have identical effects and, therefore, may not be able to distinguish pathogenic missense variants from proximal benign missense variants.

Since variant-level and gene-level prioritization methods leverage complementary signatures of negative selection, unifying the two lines of research for joint inference of variant effects and gene constraints should be beneficial. Recently, a couple of studies have tried to address this question [25–27], but all of these methods are supervised machine learning models trained on disease variants. Therefore, we lack an evolution-based statistical framework to combine predictive variant features and gene-level selective constraints for variant and gene prioritization. Based on a novel deep learning framework, *i.e.*, deep mixed-effects model, we develop **UNEECON** (**UN**ified inferenc**E** of variant **E**ffects and gene **CON**straints), an evolution-based framework to predict deleterious variants and constrained genes from both variant features and gene-level intolerance to missense mutations. By integrating 30 predictive variant features and genomic variation from 141,456 human genomes [28, 29], UNEECON outperforms existing variant effect predictors and gene constraint scores in predicting pathogenic missense variants with a dominant mode of inheritance. In addition, deleterious *de novo* variants predicted by UNEECON are strongly enriched in individuals affected by severe development disorders [30, 31], highlighting its power for interpreting the effects of *de novo* mutations. Furthermore, UNEECON provides estimates of gene-level selective constraints (UNEECON-G scores) for all protein-coding genes. In the setting of gene prioritization, UNEECON-G scores show better performance than previous gene constraint scores in predicting human essential genes [32], mouse essential genes [33, 34], autosomal dominant disease genes [35, 36], and haploinsufficient genes [37]. UNEECON is a powerful framework for both variant and gene prioritization.

## Results

### UNEECON integrates variant- and gene-level information to predict detrimental variants and constrained genes

The key idea of UNEECON is to combine variant-level predictive features, such as sequence conservation scores and protein structural features, and gene-level signatures of selective constraints to infer negative selection on every potential missense mutation in the human genome. Inspired by classical sequence conservation models [38–42], which use site-specific substitution rate as a proxy of negative selection, we utilize the relative probability of the occurrence of a potential missense mutation in human populations, compared to neutral mutations, as an allele-specific predictor of negative selection.

We first fit a mutation model to calculate $\mu_{ij}$, *i.e.*, the probability of the occurrence of mutation $i$ in gene $j$ when selection is absent. Our mutation model is trained on putatively neutral mutations in the gnomAD data [29] and captures the impact of multiple neutral and technical factors, including the 7-mer sequence context centered on the focal site, the identity of the alternative allele, the local mutation rate, and the average sequencing coverage, on the probability of occurrence of a neutral mutation. At the gene level, the number of synonymous mutations predicted by the mutation model was nearly perfectly correlated with the observed number of synonymous mutations (Spearman's $\rho = 0.976$; Fig A in S1 File), suggesting that the mutation model can serve as a proper baseline for inferring negative selection.

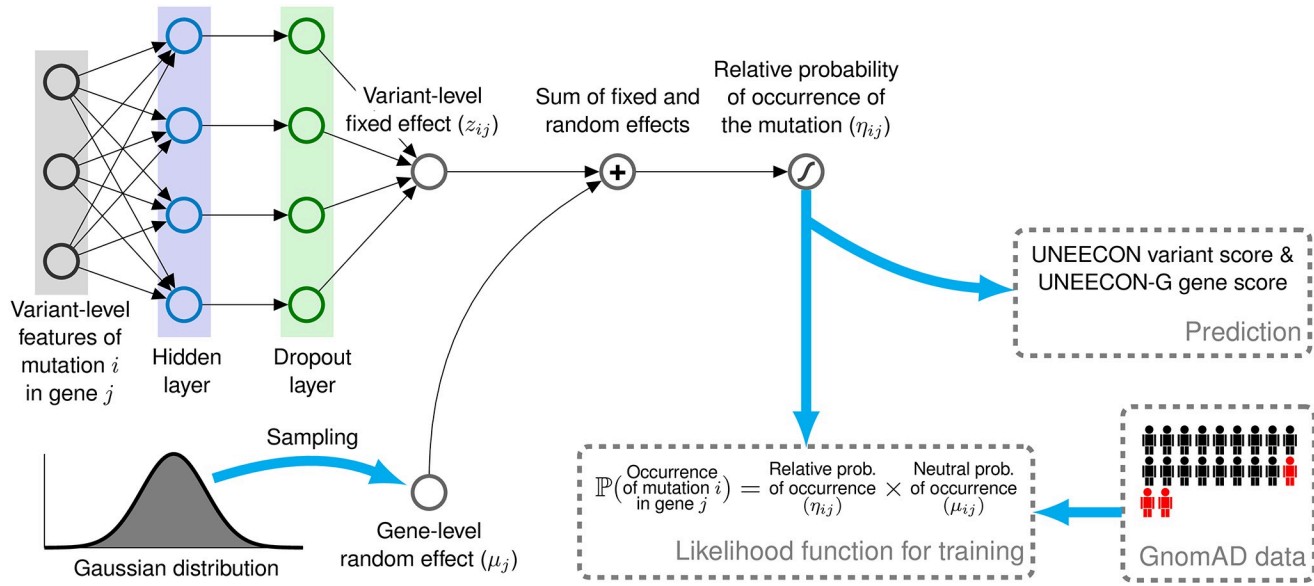

**Fig 1. Overview of the UNEECON model.** UNEECON estimates negative selection on missense mutation $i$ in gene $j$ based on the relative probability of the occurrence of the missense mutation, $\eta_{ij}$, compared to the occurrence probability of neutral mutations, $\mu_{ij}$. $\eta_{ij}$ depends on the sum of a variant-level fixed effect, $z_{ij}$, and a gene-level random effect, $u_j$. We assume that $z_{ij}$ captures the contribution of variant-level features, $\mathbf{X}_{ij}$, to negative selection, and model the relationship between $\mathbf{X}_{ij}$ and $z_{ij}$ with a feedforward neural network. We assume that $u_j$ is a Gaussian random variable modeling the gene-level variation of selective constraints that cannot be predicted from variant features. The sum of $z_{ij}$ and $u_j$ is then sent to a logistic function to obtain $\eta_{ij}$. The neutral occurrence probability, $\mu_{ij}$, is from a context-dependent mutation model trained on putatively neutral mutations. Free parameters of the UNEECON model are estimated by minimizing the discrepancy between the predicted occurrence probability, $\eta_{ij} \cdot \mu_{ij}$, and the observed occurrence of each potential missense mutation in the gnomAD exome sequencing data [29].

Given the mutation model, UNEECON employs a novel deep learning framework, deep mixed-effects model, to infer the relative probability of the occurrence of each potential missense mutation (Fig 1). In more detail, we denote $\eta_{ij}$ as the relative probability of the occurrence of missense mutation $i$ in gene $j$ with respect to the neutral occurrence probability, $\mu_{ij}$. $\eta_{ij}$ captures the impact of natural selection, instead of mutation and genetic drift, on the occurrence of a missense mutation. Analogous to generalized linear mixed models, we assume that $\eta_{ij}$ depends on the sum of a variant-level fixed effect, $z_{ij}$, and a gene-level random effect, $u_j$. We assume that $z_{ij}$ captures the contribution of predictive variant features to negative selection. Denoting $\mathbf{X}_{ij}$ as the vector of predictive features associated with mutation $i$ in gene $j$, we use a feedforward neural network to model the relationship between $\mathbf{X}_{ij}$ and $z_{ij}$ (Fig 1). Furthermore, we assume that the random-effect term, $u_j$, is a Gaussian (normal) random variable which models the variation of gene-level constraints that is not predictable from feature vector $\mathbf{X}_{ij}$. We then perform a logistic transformation on the sum of $z_{ij}$ and $u_j$ to obtain $\eta_{ij}$. Finally, we multiply $\eta_{ij}$ and $\mu_{ij}$ to obtain the occurrence probability of missense mutation $i$ in the gnomAD exome sequencing data.

We estimate the parameters of the UNEECON model by minimizing the discrepancy between the predicted occurrence probability, $\eta_{ij} \cdot \mu_{ij}$, and the observed presence/absence of each potential missense mutation, which is equivalent to maximizing a Bernoulli likelihood function. We also employ regularization techniques, including dropout [43] and early stopping [44], to avoid overfitting. After training, we calculate variant-effect scores (UNEECON scores) and gene-level intolerance to missense mutations (UNEECON-G scores) defined as the expected reduction of the occurrence probability at variant and gene levels, respectively. Both UNEECON and UNEECON-G scores range from 0 to 1, with higher scores suggesting stronger negative selection.

## UNEECON scores capture variation of negative selection within and across genes

We trained the UNEECON model with 30 missense variant features, including conservation scores, protein structural features, and regulatory features (Table A in S1 File), and rare missense variants with a minor allele frequency (MAF) lower than 0.1% in the gnomAD dataset [29]. After the training process, we first compared the distributions of UNEECON scores across potential missense mutations in haploinsufficient genes [37], autosomal dominant disease genes [35, 36], autosomal recessive disease genes [35, 36], and olfactory receptor genes [45]. In agreement with previous studies [16, 28, 35], missense mutations in autosomal dominant disease genes had higher UNEECON scores than those in autosomal recessive disease genes (Fig 2a), suggesting a stronger selection on heterozygous missense mutations in the genes associated with autosomal dominant disorders. Also, we observed that missense mutations in olfactory receptor genes had much lower UNEECON scores (Fig 2a), potentially due to relaxed purifying selection or enhanced positive selection in the human lineage [46].

Second, we investigated whether the distributions of UNEECON scores varied across different types of protein regions annotated by UniProt [47, 49] and MobiDB [48]. We observed that the distributions of UNEECON scores were similar among α-helices, β-strands, and hydrogen-bonded turns (Fig 2b). In contrast, UNEECON scores were significantly lower in disordered protein regions. As expected, UNEECON scores were significantly higher in functional protein sites, including enzyme active sites and ligand binding sites (Fig 2b), suggesting much stronger purifying selection on these critical residues.

Interestingly, UNEECON scores showed a bimodal distribution in both enzyme active sites and ligand binding sites, with the first mode around 0.8 and the second mode around 0.2 (Fig 2b). Therefore, even though most of enzyme active sites and ligand binding sites are believed to play a crucial role in maintaining the functional integrity of proteins, a substantial fraction of heterozygous missense mutations in these sites are not subject to strong negative selection. This result may be due to the fact that heterozygous mutations in recessive genes have a limited impact on protein function. In agreement with this explanation, functional protein sites in autosomal recessive disease genes had substantially lower UNEECON scores than those in haploinsufficient and autosomal dominant genes (Fig B in S1 File).

Furthermore, we used the human CDKL5 protein as an example to illustrate UNEECON's capability to capture regional variation of negative selection within a gene. Similar to a previously published metric of regional missense constraints [25], UNEECON predicted that the N-terminus of CDKL5 was under strong negative selection (Fig 2c). In agreement with the evidence of strong selection, this region was also enriched with known pathogenic missense variants and depleted with benign missense variants (Fig 2c). Interestingly, UNEECON also predicted that the C-terminus of CDKL5 was under much weaker selection compared with the rest of the protein (Fig 2c). Accordingly, this unconstrained region was enriched with benign missense variants and depleted with pathogenic missense variants.

## UNEECON scores accurately predict missense mutations associated with dominant genetic disorders

Since UNEECON measures the strength of strong negative selection on heterozygous missense mutations, we hypothesize that UNEECON scores are predictive of missense variants associated with dominant Mendelian disorders. To test this hypothesis, we compared the performance of UNEECON with eight existing methods in the setting of predicting autosomal dominant disease variants in ClinVar [30]. The eight previously published methods can be classified into three categories: 1) variant-level prediction methods, including LASSIE [12],

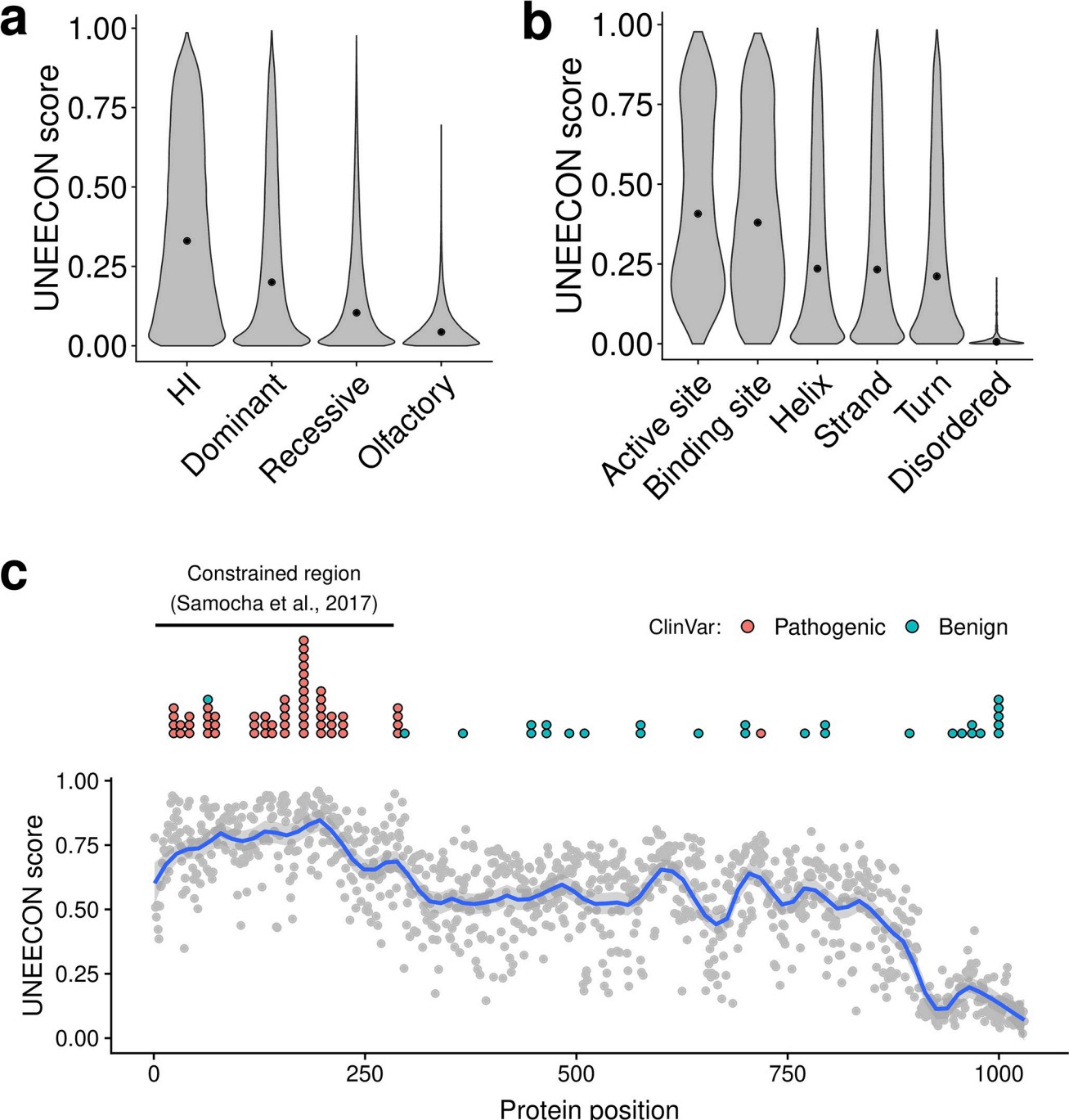

**Fig 2. Distributions of UNEECON scores across potential missense mutations.** (a) Distributions of UNEECON scores estimated for potential missense mutations in haploinsufficient (HI) genes [37], autosomal dominant disease genes [35, 36], autosomal recessive disease genes [35, 36], and olfactory receptor genes [45]. (b) Distributions of UNEECON scores estimated for potential missense mutations in various protein regions. The functional sites and protein secondary structures are based on UniProt annotations [47]. The predicted disordered protein regions are from MobiDB [48]. (c) Average UNEECON scores estimated for all codon positions in the CDKL5 protein. Each grey dot represents the UNEECON score averaged over all missense mutations in a codon position. Blue curve represents the locally estimated scatterplot smoothing (LOESS) fit. Blue and red dots represent pathogenic and benign missense variants from ClinVar [30], respectively. The horizontal line represents a constrained region reported in a previous study [25].

CADD [8], Eigen [50], and PrimateAI [11]; 2) gene-level and region-level prediction methods, including RVIS [16], pLI [17], and CCR [22]; and 3) MPC [25], a supervised machine learning method utilized both variant features and regional constraint scores as input features. It is worth noting that several aforementioned methods, including UNEECON, used the variants from the gnomAD dataset as a part of their training data. If the same gnomAD variants are also used as negative controls in the evaluation of performance, we will overestimate the power of these methods trained on the gnomAD data. To avoid this problem, we used benign missense variants in ClinVar as negative controls and removed any ClinVar variants that are also present in gnomAD. Then, we matched the numbers of positives and negatives by random sampling without replacement.

Overall, UNEECON outperformed previous methods in predicting ClinVar missense variants associated with autosomal dominant disorders (Fig 3a; Table B in S1 File). Among the previously published methods, variant-level predictors performed better than gene-level and region-level constraint scores. To test the robustness of these results, we constructed an alternative set of dominant pathogenic variants defined as all the pathogenic missense variants located in 709 genes associated with autosomal dominant diseases [35, 36]. UNEECON again outperformed the other methods in this dataset, even though the difference in performance between UNEECON and LASSIE was not statistically significant (Fig C in S1 File; Table B in S1 File).

Because some genes are better studied than others in medical genetics, known pathogenic variants are not evenly distributed across genes [51]. In the ClinVar database, the majority of protein-coding genes contain only a single class of variants, *i.e.*, either "pathogenic-only" or "benign-only". It is considerably more challenging to predict pathogenic variants in "mixed" genes that contain both pathogenic and benign variants [51]. We evaluated UNEECON on 157 autosomal dominant genes containing both pathogenic and benign variants. UNEECON again outperformed previous methods in separating pathogenic missense variants from benign missense variants in these "mixed" genes (Fig D1 in S1 File).

To evaluate the performance of UNEECON when the information of gene-level selective constraints is absent, we removed all protein-coding genes with at least one ClinVar pathogenic missense variant from the training data and retrained the UNEECON model on the new training set. Then, we evaluated the performance of this version of UNEECON in separating pathogenic missense variants from benign missense variants in the "mixed" genes containing both pathogenic and benign missense variants. It is worth noting that the "mixed" genes were not used in the training step. In the step of prediction, UNEECON effectively substituted the gene-level random-effect term with its genome-wide average, forcing UNEECON to make predictions solely based on variant-level features. Again, UNEECON outperformed previous methods in this setting (Fig D2 in S1 File), suggesting that UNEECON is a robust method and can still predict pathogenic variants when the information of gene-level selective constraints is absent.

While UNEECON was highly powerful in predicting autosomal dominant disease variants, it might not be as accurate when applied to predict recessive disease variants which are only detrimental in the homozygous state. To test this hypothesis, we evaluated UNEECON's performance in predicting autosomal recessive disease variants in ClinVar [30]. As expected, UNEECON was outperformed by multiple methods, such as LASSIE [12] and Eigen [50], in this setting (Fig E in S1 File). Similar results were reached with an alternative set of recessive pathogenic variants defined as all the pathogenic missense variants located in 1,183 autosomal recessive genes [35, 36] (Fig F in S1 File).

As an orthogonal benchmark of UNEECON's performance in predicting dominant disease variants, we investigated whether UNEECON was able to predict *de novo* missense mutations

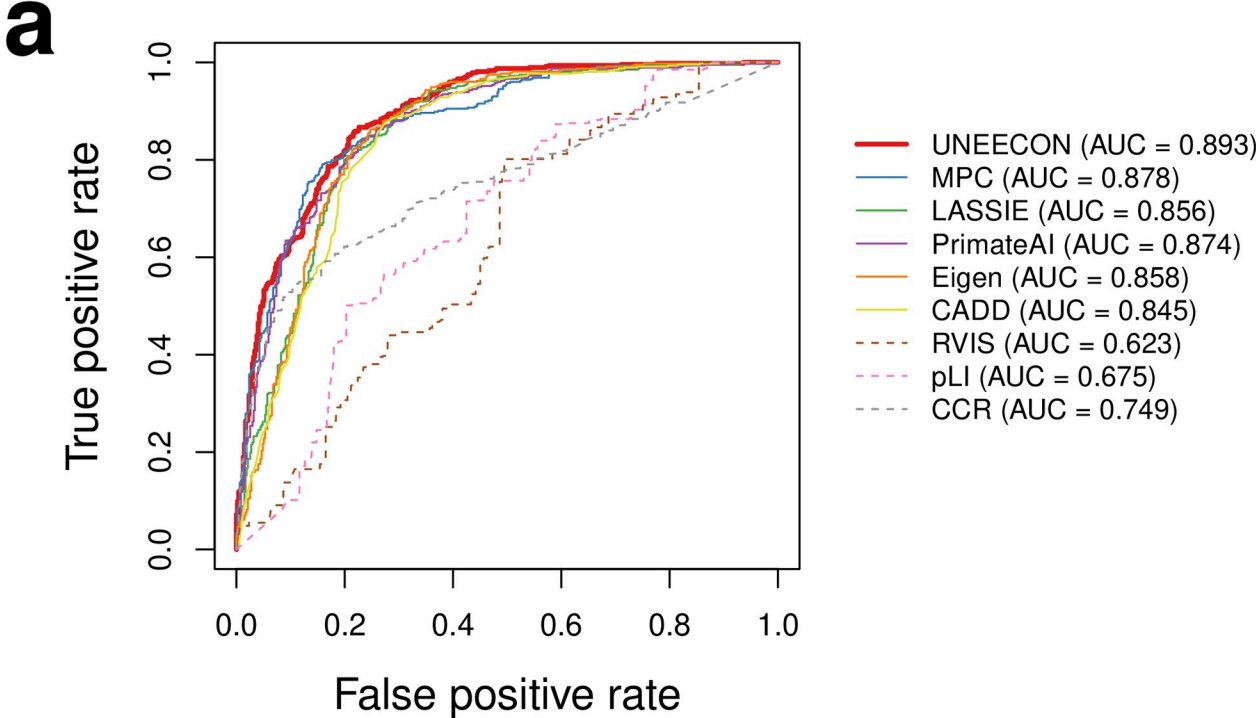

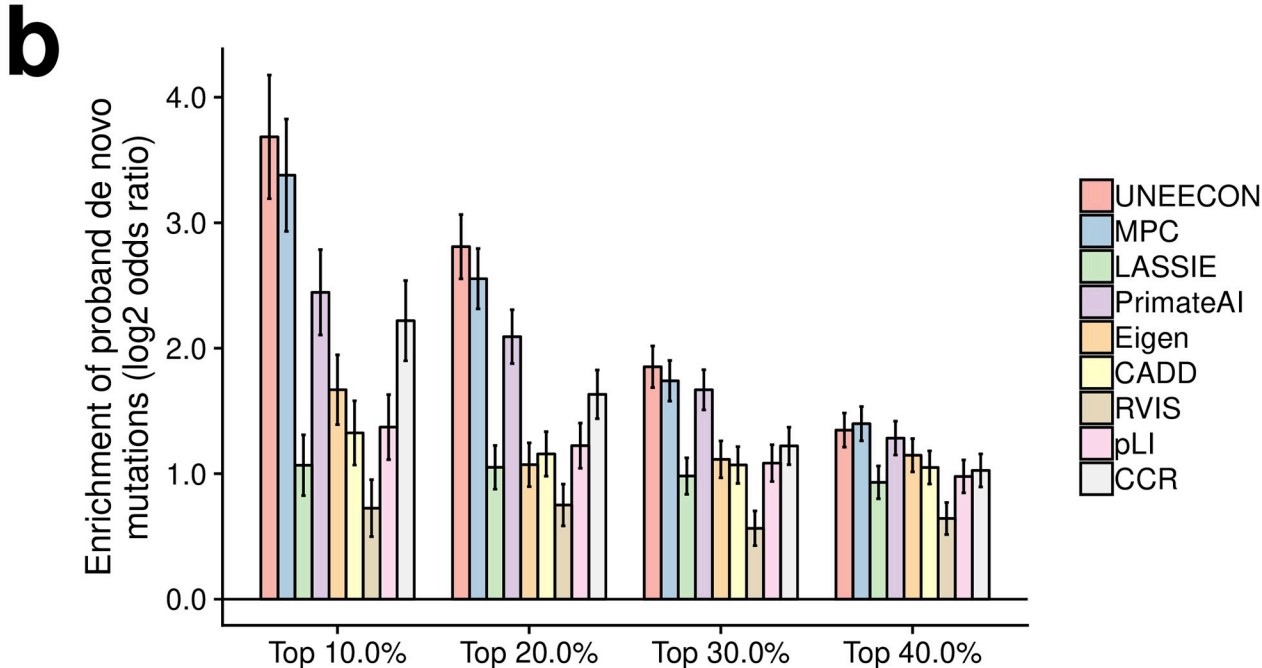

**Fig 3. Predictive power of various methods for distinguishing pathogenic missense variants from benign missense variants. (a)** Performance in predicting autosomal dominant pathogenic variants from ClinVar [30]. True positive and true negative rates correspond to the fractions of pathogenic and benign variants exceeding various thresholds, respectively. AUC corresponds to the area under the receiver operating characteristic curve. **(b)** Enrichment of predicted deleterious *de novo* variants in individuals affected by developmental disorders [31]. The *y*-axis corresponds to the $\log_2$ odds ratio of the enrichment of predicted deleterious variants in the affected individuals for a given percentile threshold. The *x*-axis corresponds to the various percentile threshold values used in the enrichment analysis. Error bars represent the standard error of the $\log_2$ odds ratio.

identified in individuals affected by severe developmental disorders [31]. We obtained *de novo* missense variants identified in affected individuals and healthy individuals from denovo-db [52]. Then, for multiple percentile rank cutoff values (top 10%, 20%, 30%, and 40%), we evaluated the enrichment of deleterious variants predicted by each method in the affected individuals. Overall, missense variants predicted by UNEECON and MPC showed the highest enrichments in the affected individuals (Fig 3b), suggesting that these two methods were more powerful in predicting *de novo* risk mutations. The performance gaps between UNEECON/ MPC and the other methods were highest at the most stringent cutoff of 10% (Fig 3b).

UNEECON uses a data-driven approach to combine variant-level features and gene-level selective constraints. Alternatively, variant-level and gene-level predictors can be used as two successive filters in the same variant prioritization pipeline. We compared UNEECON with such a heuristic method [16] in the setting of predicting ClinVar missense variants and *de novo* missense mutations associated with severe developmental disorders. The heuristic method converted RVIS and PolyPhen-2 scores into two binary predictors, and only the variants predicted by both RVIS and PolyPhen-2 were considered to be deleterious. Compared with RVIS and the heuristic method combining RVIS and PolyPhen-2, UNEECON showed significantly better performance in predicting pathogenic missense variants in ClinVar, and the *de novo* mutations predicted by UNEECON showed substantially stronger enrichment in the individuals affected by severe developmental disorders (Fig G in S1 File).

## UNEECON-G scores perform favorably in predicting disease genes and essential genes

The UNEECON model also provided UNEECON-G scores which measure gene-level intolerance to missense mutations. We compared the performance of UNEECON-G scores with alternative gene constraint scores, including RVIS [16], pLI [17], mis-z [17], and GDI [19], in the setting of predicting disease genes and essential genes. The disease gene sets included 709 autosomal dominant disease genes [35, 36] and 294 haploinsufficient genes [37]. The essential gene sets included 2,454 human orthologs of mouse essential genes [33, 34] and 683 human essential genes identified by CRISPR knockout experiments in human cell lines [32]. For each set of autosomal dominant disease genes, haploinsufficient genes, and mouse essential genes, we constructed a negative gene set by sampling a matched number of genes from the other genes in the human genome. For the essential genes identified by CRISPR in cell lines, we constructed a negative gene set by sampling a matched number of genes from the nonessential genes reported in the same study [32]. Overall, UNEECON-G significantly outperformed alternative gene-level metrics in predicting the four sets of essential and disease genes (Fig 4; Table C in S1 File).

## Both variant-level features and gene-level constraints are important for variant interpretation

To gain more insights into which components of the UNEECON model are critical for predicting detrimental variants, we characterized the contributions of the 30 variant features and the gene-level random effect to negative selection. Unfortunately, it is challenging to directly interpret the UNEECON model due to the nonlinearity introduced by the feedforward artificial neural network. To bypass this difficulty, we used a linear version of the UNEECON model as an approximation of the nonlinear UNEECON model. The linear UNEECON model has no hidden units and can be interpreted as a generalized linear mixed model. Then, we evaluated the importance of the variant-level features and the gene-level random effect in this linear surrogate model. The variant score predicted by the linear UNEECON model was nearly

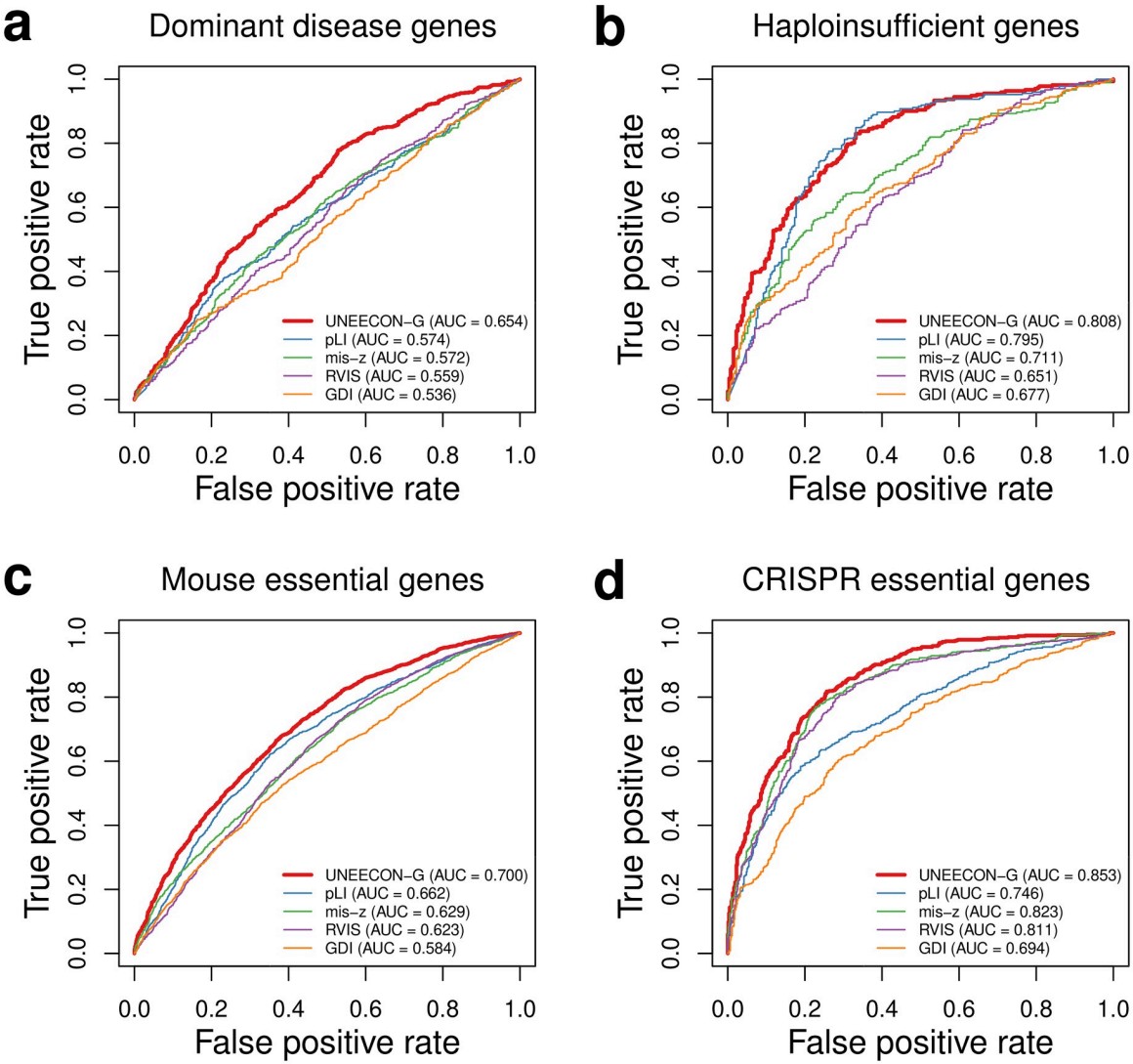

**Fig 4. Predictive power of various methods for distinguishing disease and essential genes from genes not likely to have strong phenotypic effects. (a)** Performance in predicting autosomal dominant disease genes [35, 36]. **(b)** Performance in predicting haploinsufficient genes [37]. **(c)** Performance in predicting human orthologs of mouse essential genes [33, 34]. **(d)** Performance in predicting human essential genes in cell lines [32]. True positive and true negative rates correspond to the fractions of positive and negative genes exceeding various thresholds, respectively. AUC corresponds to the area under the receiver operating characteristic curve.

perfectly correlated with the UNEECON score predicted by the original nonlinear UNEECON model (Spearman's $\rho$ = 0.95), supporting the use of the linear UNEECON model as a surrogate for model interpretation.

In the linear surrogate model, the relative probability of the occurrence of mutation $i$ in gene $j$, $\eta_{ij}$, is a linear combination of variant features, $\mathbf{X}_{ij}$, and the gene-level random effect, $u_j$, up to a logistic transformation. Analogous to the interpretation of canonical linear models, we defined the contribution score of a variant feature as the negative value of the weight (regression coefficient) associated with this feature, and similarly defined the contribution score of the gene-level random effect as the negative value of its standard deviation. Positive

and negative contribution scores suggest that the corresponding variables are positively and negatively correlated with negative selection, respectively, and contribution scores near zero indicate that the corresponding variables are not important for predicting detrimental mutations.

As shown in Fig H in S1 File, the gene-level random effect was the most important feature for predicting deleterious variants, highlighting the importance of gene-level constraints for variant interpretation. In addition, several conservation scores, such as the qualitative predictions from LRT, PROVEAN, and SIFT, and a subset of protein structural features, such as the predicted probabilities of forming various protein secondary structures, were also important for accurate prediction of deleterious variants. Even though these variant features individually were less important than the gene-level random effect, accumulatively they explained a significant fraction of the variation of negative selection across missense mutations. Therefore, both variant features and gene-level constraints are important for predicting deleterious variants.

## Missense constraints and loss-of-function constraints are weakly correlated across genes

The UNEECON-G score represents gene-level intolerance to missense mutations in human populations. Alternatively, selective constraint on a gene can be defined as its degree of intolerance to loss-of-function mutations, such as stop-gained, frameshift, and splice-site mutations [17, 28, 29]. A recent study suggested that gene-level intolerance to loss-of-function mutations may not be a reliable predictor of pathogenic missense variants [53], implying a weak correlation between gene-level intolerance to loss-of-function mutations and that to missense mutations. To test this hypothesis at a genome-wide scale, we investigated the correlation between the UNEECON-G score and the pLI score, a metric of gene-level intolerance to loss-of-function mutations [17]. As expected, the UNEECON-G score was only moderately correlated with the pLI score (Spearman's $\rho$ = 0.59; Fig 5a).

Intrinsically disordered proteins play key roles in a multitude of biological processes [54], but their sequences are poorly conserved across species [55]. Therefore, missense mutations in intrinsically disordered proteins may be under weak negative selection even if loss-of-function mutations in these proteins are deleterious. To test this hypothesis, we investigated the distributions of protein disorder content, *i.e.*, the fraction of disordered regions in a protein [48], across 1,912 protein-coding genes intolerant to heterozygous loss-of-function mutations (pLI score $\geq$ 0.9). We split the 1,912 genes into two equal-size groups based on their UNEECON-G scores, and defined the 956 genes with higher UNEECON-G scores as the gene set intolerant to both missense and loss-of-function mutations. Accordingly, we defined the 956 genes with lower UNEECON-G scores as the gene set tolerant to missense but not to loss-of-function mutations. Compared with the 956 genes intolerant to both missense and loss-of-function mutations, the disorder contents were significantly higher in the 956 genes tolerant to missense mutations but not to loss-of-function mutations (Fig 5b). Therefore, disordered protein regions can at least partially explain the observed discrepancy between missense constraints and loss-of-function constraints.

We further investigated the enrichment of Reactome pathways [56] and Gene Ontology terms [57, 58] in the 956 genes intolerant to both missense and loss-of-function mutations, using the 956 genes tolerant to missense but not to loss-of-function mutations as a background. Several key pathways associated with central nervous system, gene regulation, cell cycle, and innate immune response were overrepresented in the genes intolerant to both

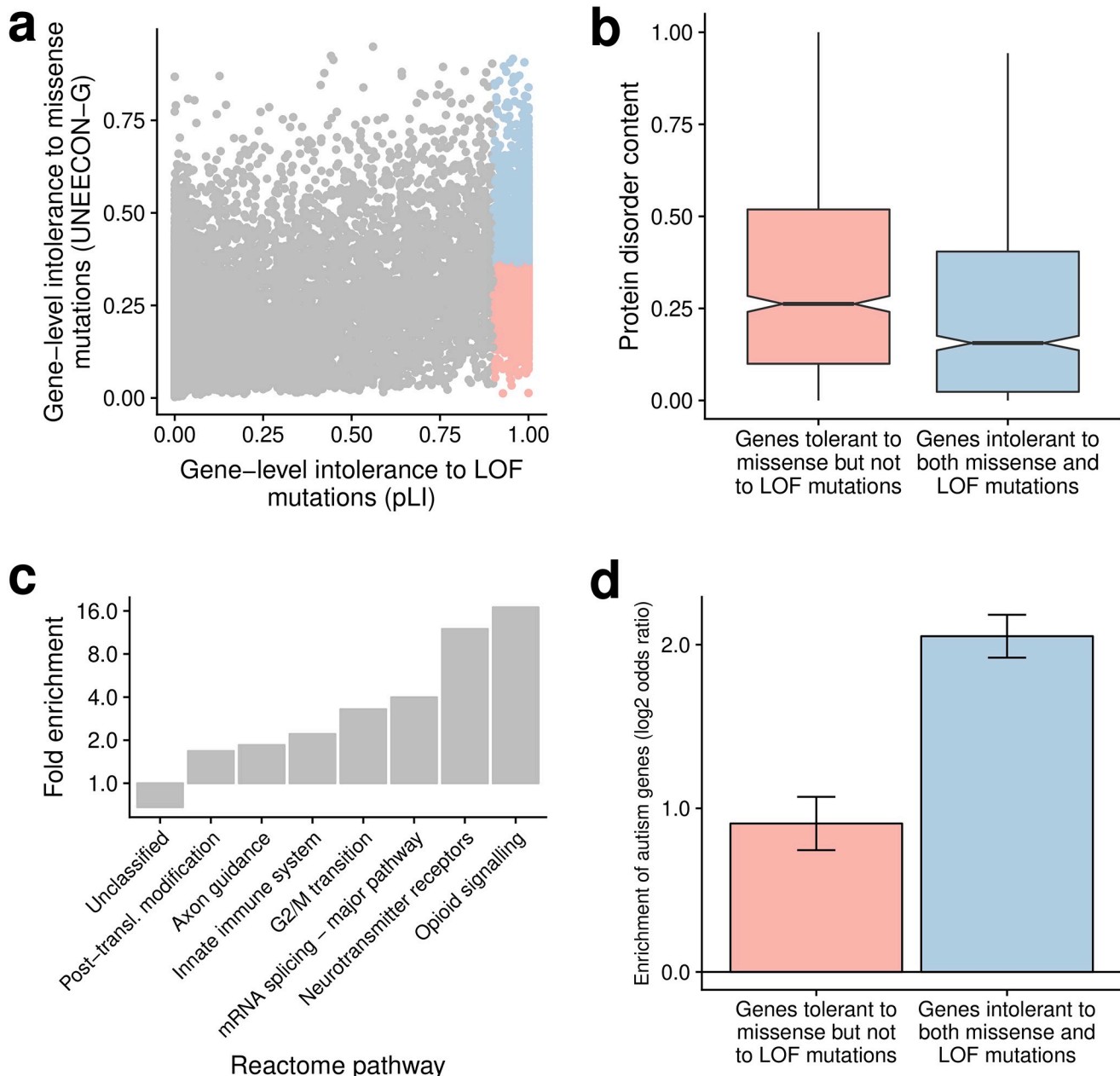

**Fig 5. Distributions of gene-level intolerance to missense and to loss-of-function mutations. (a)** Correlation between gene-level intolerance to missense mutations (UNEECON-G score) and that to loss-of-function (LOF) mutations (pLI score). Blue dots represent 956 genes intolerant to both missense and LOF mutations. Red dots represent 956 genes tolerant to missense but not to loss-of-function mutations. **(b)** Distribution of protein disorder content in the gene sets intolerant to loss-of-function mutations. **(c)** Enrichment of Reactome pathways in the gene set intolerant to both missense and loss-of-function mutations. The gene set tolerant to missense but not to loss-of-function mutations is used as a background. Only the highest-level Reactome terms from the PANTHER hierarchy view are included in the visualization. The term "unclassified" indicates that the corresponding genes have no known or inferred function. A fold enrichment below 1 indicates a depletion in the gene set intolerant to both missense and loss-of-function mutations, or equivalently, an enrichment in the gene set tolerant to missense but not to loss-of-function mutations. **(d)** Enrichment of autism genes in the gene sets intolerant to loss-of-function mutations. Error bars represent the standard error of the $\log_2$ odds ratio.

missense and loss-of-function mutations (Fig 5c; Table D in S1 File). We observed an enrichment of similar terms based on the Gene Ontology (Tables E & F in S1 File). In agreement with the enrichment of pathways associated with the central nervous system, the genes intolerant to both missense and loss-of-function mutations were strongly enriched in the gene set

implicated in the autism spectrum disorders [59] (Fig 5d). In contrast, the genes tolerant to missense but not to loss-of-function mutations were more likely to have no known or inferred function (Fig 5c; Table D in S1 File).

## Discussion

Here we present UNEECON for a unified prediction of deleterious missense mutations and highly constrained genes in the human genome. Compared with previously published methods, UNEECON shows superior performance in predicting missense variants and protein-coding genes associated with dominant genetic disorders. Therefore, UNEECON is a promising framework for both variant and gene prioritization. Furthermore, unlike supervised machine learning approaches, such as MPC [25], UNEECON integrates variant features and gene constraints based on signatures of negative selection instead of labeled disease variants. Therefore, UNEECON is unlikely to suffer from the circularity and the inflated performance commonly found in supervised methods [51].

It is worth noting that UNEECON is different from existing prioritization methods in multiple important aspects. First, unlike classical sequence conservation metrics [38–42] and integrative variant scores [8–15], UNEECON infers the strength of negative selection based on rare genetic variants. Therefore, UNEECON predicts strongly deleterious variants with a dominant/semidominant mode of inheritance [60]. Second, unlike previously published gene constraint metrics that typically assign the same score to all missense variants within a gene [16–23], UNEECON assigns different scores to the variants in the same gene in a feature-dependent manner, providing high-resolution maps of variant effects within genes. Third, UNEECON is able to adjust the distribution of variant scores within a gene according to the degree of depletion of missense variants in this gene, allowing for assigning different scores to missense mutations with similar variant features but located in different genes.

Similar to other metrics of negative selection, the performance of UNEECON strongly depends on the correlation between variant penetrance and negative selection. Our analysis of pathogenic variants suggests that UNEECON can predict deleterious variants associated with dominant genetic disorders but not necessarily those associated with recessive disorders. Also, because common variants are unlikely to be under strong negative selection [12], UNEECON scores might not be able to predict common variants associated with complex traits or late-onset diseases. In contrast, a recent study suggests that rare variants associated with complex traits are strongly enriched in coding regions and tend to be under negative selection [61], implying that UNEECON could be a useful tool for rare variant association studies.

UNEECON is based on a novel machine learning framework, deep mixed-effects model, to integrate variant features and gene-level constraints. By comparing the nonlinear UNEECON model with a linear UNEECON model without hidden layers, we observe that the UNEECON scores from the nonlinear model are nearly perfectly correlated with the scores from the linear UNEECON model (Spearman's $\rho$ = 0.95). Therefore, the additional nonlinearity introduced by neural networks may not be critical for the dataset described in this work. Nevertheless, the deep mixed-effects model has the flexibility of modeling complex interactions between variant-level features, which may be important for analyzing other datasets.

Our analysis of the linear UNEECON model suggests that both the predictive variant features and the gene-level constraints are important for variant interpretation. In particular, gene-level evidence of selective constraints is the single most important predictor of negative selection on missense mutations. Also, predictive variant features accumulatively explain a large fraction of the variation of negative selection across missense variants. We expect that the

combination of variant-level predictors and gene-level constraints will be an essential component in the future development of variant and gene prioritization methods.

By contrasting UNEECON-G scores against pLI scores, we observe a low correlation between gene-level intolerance to missense mutations and that to loss-of-function mutations. The prevalence of disordered protein regions in the human proteome is a key biological factor contributing to the low correlation between missense and loss-of-function constraints. Furthermore, we observe that the genes intolerant to both loss-of-function mutations and missense mutations may play key roles in the central nervous system and the autism spectrum disorders, highlighting the needs to investigate the function of these genes using state-of-the-art experimental techniques. By combing powerful variant and gene prioritization tools, such as UNEECON, and high-throughput mutagenesis and genome editing techniques [62, 63], we will obtain more insights into the function of these strongly constrained genes in the future.

## Materials and methods

### Predictive variant features

UNEECON was trained on 30 missense variant features previously used to infer fitness effects of coding variants in the human genome (Table A in S1 File; [12]). These features can be classified into three categories: sequence conservation scores, protein structural features, and functional genomic features. The sequence conservation scores included SIFT [5], PROVEAN [7], SLR [64], Grantham [65], PSIC [66], LRT [67], MutationAssessor [4], HMMEntropy [68], and phyloP scores [40]. The protein structural features included predicted secondary structures, B-factors, contributions to protein stability, and relative solvent accessibilities from SNVBox [68]. The functional genomic features included the non-commercial version of SPIDEX splicing scores [69] and the maximum RNA-seq signals from the Roadmap Epigenomics Project [70]. Following a common practice in machine learning and statistics, we standardized continuous features by subtracting the mean and dividing by the standard deviation. All the features were based on the hg19 (GRCh37) assembly.

### Population genomic data

We dowloaded whole genome variation data, exome variation data, and corresponding sequencing coverage data from the gnomAD browser [28, 29] (version 2.1.0). We only retained rare SNVs (MAF < 0.1%) that passed gnomAD's built-in quality filter for downstream analysis.

### Context-dependent mutation model

We trained a context-dependent mutation model to capture the impact of 7-mer sequence context, local mutation rate, and sequencing coverage on the probability of the occurrence of each mutation in the gnomAD exome sequencing data. Because of the intrinsic sparsity of putatively neutral variants in coding regions, it is difficult to build a mutation model solely based on the gnomAD exome sequencing data. Therefore, we first built a mutation model based on neutral noncoding variants in the gnomAD whole genome sequencing (WGS) data. Then, we recalibrated the WGS-based mutation model in the gnomAD exome sequencing data to adjust for the differences in population sample size and sequencing coverage between the WGS and the exome sequencing data.

To build the WGS-based mutation model, we first compiled a list of putatively neutral noncoding regions following a strategy described in previous studies [10, 71, 72]. We removed coding exons [73], conserved phastCons elements [74], nucleotide sites within 1000 bp of any

coding exons, and nucleotide sites within 100 bp of any phastCons elements. We assumed that the remaining noncoding regions were largely depleted with functional elements and, therefore, mutations in these regions were putatively neutral. It is worth noting that we also removed any nucleotide sites with an average sequencing coverage below 20, due to the difficulty of variant calling in low coverage regions, and any sites overlapping CpGs, due to the high mutation rates in these sites.

Then, we fit a WGS-based mutation model to the gnomAD WGS data in putatively neutral noncoding regions. First, for each possible combination of mutation $l$ and sequence context $k$ (the focal nucleotide with 3 flanking nucleotides on each side), we calculated its mutability, $f_{k \to l}$, defined as the proportion of 7-mer sequences of $k$ with observed rare variant $l$ in the gnomAD WGS data. We defined $F_{k \to l} = \mathrm{logit}(f_{k \to l}) \equiv \log\left(\frac{f_{k \to l}}{1 - f_{k \to l}}\right)$, which represents the contribution of sequence context and mutation type to the probability of variant occurrence in logit scale. Second, we fit a genome-wide logistic regression model to estimate the contributions of sequencing depth and context-dependent mutability to the probability of the occurrence of a mutation. This logistic regression model assumed

$$P(Y_i^{\mathrm{WGS}} = 1) = \mathrm{logistic}(\alpha_0 + \alpha_1 \log(d_i) + \alpha_2 F_{k_i \to l_i}), \tag{1}$$

where $Y_i^{\mathrm{WGS}}$ is a binary indicator of the occurrence of neutral noncoding mutation $i$ in the gnomAD WGS data. $\alpha_0$, $\alpha_1$, and $\alpha_2$ are the free parameters in the logistic regression model. $d_i$ is the average sequencing depth at the nucleotide position of mutation $i$. $k_i$ and $l_i$ are the sequence context and the mutation type of mutation $i$, respectively. We then fit a local logistic regression model for each exon $m$ in the human genome with

$$P(Y_i^{\mathrm{WGS}} = 1) = \mathrm{logistic}(\alpha_3^m + \hat{\alpha}_1 \log(d_i) + \hat{\alpha}_2 F_{k_i \to l_i}), \quad \text{for all mutation } i\text{'s}$$
$$\text{within 60 kb of exon } m \tag{2}$$

where $\alpha_3^m$ is an exon-specific free parameter independently estimated for each exon $m$, and $\hat{\alpha}_1$ and $\hat{\alpha}_2$ are the estimates of regression coefficients in Eq 1. The local regression model effectively added an exon-specific, multiplicative scaling factor to adjust for the variation of local mutation rates across exons. We defined

$$q_i = \hat{\alpha}_3^m + \hat{\alpha}_1 \log(d_i) + \hat{\alpha}_2 F_{k_i \to l_i} \tag{3}$$

as the logit of the predicted occurrence probability of mutation $i$ in exon $m$ in the WGS-based mutation model, given the estimated $\hat{\alpha}_3^m$, $\hat{\alpha}_1$, and $\hat{\alpha}_2$.

Finally, we recalibrated the WGS-based mutation model in the gnomAD exome sequencing data with a logistic regression model,

$$P(Y_i^{\mathrm{exome}} = 1) = \mathrm{logistic}(\beta_0 + q_i), \tag{4}$$

where $Y_i^{\mathrm{exome}}$ is a binary indicator of the presence of synonymous mutation $i$ in the gnomAD exome sequencing data, and $\beta_0$ is the free parameter of the logistic regression model. The exome-based logistic regression model effectively added a multiplicative scaling factor to accommodate for the differences in population sample size and sequencing coverage between the WGS and the exome sequencing data. In the final exome mutation model, we defined the predicted probability of the occurrence of missense mutation $i$ in gene $j$ as

$$\mu_{ij} = \mathrm{logistic}(\hat{\beta}_0 + q_i) \equiv \frac{\exp(\hat{\beta}_0 + q_i)}{1 + \exp(\hat{\beta}_0 + q_i)}, \tag{5}$$

where $\hat{\beta}_0$ is the maximum likelihood estimate of $\beta_0$ in Eq 4. We fit all the logistic regression models using the glm function in R [75].

## Details of the UNEECON model

We assume that negative selection on a potential mutation is a mathematical function of the sum of a variant-level fixed effect, $z_{ij}$, and a gene-level random effect, $u_j$ (Fig 1). Denoting $\mathbf{X}_{ij}$ as the vector of variant features associated with mutation $i$ in gene $j$, we assume that the variant-level fixed effect, $z_{ij}$, can be modeled by a feedforward neural network. At the bottom of the neural network, a nonlinear hidden layer and a dropout layer are employed to transform $\mathbf{X}_{ij}$ to a hidden vector,

$$\mathbf{H}_{ij} = \mathrm{droput}(\mathrm{ReLU}(\mathbf{X}_{ij} \cdot \mathbf{W}_{\mathrm{hidden}} + \mathbf{B}_{\mathrm{hidden}})), \tag{6}$$

where **ReLU** and **dropout** are the rectified linear layer [76] and the dropout layer [43], respectively. $\mathbf{W}_{\mathrm{hidden}}$ and $\mathbf{B}_{\mathrm{hidden}}$ are the weight matrix and bias vector of the hidden layer, respectively. Then, we assume the fixed effect term, $z_{ij}$, is a linear combination of the hidden vector $\mathbf{H}_{ij}$,

$$z_{ij} = \mathbf{H}_{ij} \cdot \mathbf{W}_{\mathrm{output}} + b_{\mathrm{output}}, \tag{7}$$

where $\mathbf{W}_{\mathrm{output}}$ and $b_{\mathrm{output}}$ are the weight vector and the bias term, respectively. We use the Glorot method [77] to initialize the weights, $\mathbf{W}_{\mathrm{hidden}}$ and $\mathbf{W}_{\mathrm{output}}$, and initialize the bias terms, $\mathbf{B}_{\mathrm{hidden}}$ and $b_{\mathrm{output}}$, with zeros. In the dropout layer, we use a fixed dropout rate of 0.5. It is worth noting that, if a linear version of the UNEECON model is used, Eq 6 will be replaced by an identity function $\mathbf{H}_{ij} \equiv \mathbf{X}_{ij}$.

To capture the variation of gene-level selective constraints that cannot be predicted from feature vector $\mathbf{X}_{ij}$, we introduce a gene-level random-effect term following a Gaussian distribution,

$$u_j \sim \mathcal{N}(0, \sigma), \tag{8}$$

where $\sigma$ is the standard deviation of the Gaussian distribution. Given the fixed and random effect terms, we assume $\eta_{ij}$, the relative probability of the occurrence of mutation $i$ in gene $j$, follows

$$\eta_{ij} = \mathrm{logistic}(z_{ij} + u_j) \equiv \frac{\exp(z_{ij} + u_j)}{1 + \exp(z_{ij} + u_j)}. \tag{9}$$

We further assume that the total probability of the occurrence of potential missense mutation $i$ in gene $j$ is equal to the product of $\eta_{ij}$ and $\mu_{ij}$, the probability of variant occurrence under the neutral mutation model described in Eq 5. Accordingly, the likelihood function for the data associated with gene $j$ is defined as

$$\mathcal{L}_j = \prod_i (\eta_{ij}\mu_{ij})^{Y_{ij}}(1 - \eta_{ij}\mu_{ij})^{1-Y_{ij}}, \tag{10}$$

where $Y_{ij}$ is a binary indicator of the occurrence of missense mutation $i$ in gene $j$ in the gnomAD exome sequencing data.

## Training the UNEECON model

We trained UNEECON with the Adam optimization algorithm [78]. In each iteration, UNEECON loaded a mini-batch of data consisting of all the potential missense mutations in a single

gene. Then, we calculated the log likelihood of the mini-batch of data using Eq 10 and numerically integrated out $u_j$ with a 20-point Gaussian Quadrature rule. The negative log value of Eq 10 was used as the objective function in the Adam optimizer.

We randomly split the data into a training set (80% genes; 51,108,443 potential missense mutations), a validation set (10% genes; 6,435,990 potential missense mutations), and a test set (10% genes; 6,414,968 potential missense mutations). After each epoch of training, UNEECON evaluated the objective function in the validation set and stopped training when the objective function did not improve over 5 successive epochs to avoid overfitting (early stopping). Furthermore, we performed a grid search to optimize two hyperparameters, *i.e.*, the learning rate of the Adam algorithm ($10^{-2}$, $10^{-3}$, and $10^{-4}$) and the number of hidden units (64, 128, 256, 512, and a linear model without hidden units). The model was evaluated on the test dataset to choose optimal hyperparameters. We observed that a nonlinear UNEECON model with 512 hidden units and a learining rate of $10^{-4}$ had the lowest negative log likelihood on the test dataset. Therefore, we chose this model as the optimal one for downstream analysis.

## Calculating UNEECON and UNEECON-G scores

After training, we fixed the free parameters, *i.e.*, $\mathbf{W}_{\text{hidden}}$, $\mathbf{W}_{\text{output}}$, $\mathbf{B}_{\text{hidden}}$, $b_{\text{output}}$, and $\sigma$, to the estimated values in the optimal model. Then, we calculated the UNEECON score for all the potential missense mutations in the human genome. For each gene $j$, UNEECON first calculated the posterior distribution of the random effect, $\mathbb{P}(u_j | \text{Data}_j)$, where $\text{Data}_j = \{\mathbf{X}_{1j}, \mathbf{X}_{2j}, \cdots, \mathbf{X}_{Nj}, Y_{1j}, Y_{2j}, \cdots, Y_{Nj}\}$ are the features and indicators of occurrence of all the potential missense mutations in gene $j$. Denoting $\mathbb{E}_{u_j | \text{Data}_j}(\eta_{ij} | u_j) \equiv \int \eta_{ij} \mathbb{P}(u_j | \text{Data}_j) du_j$ as expected relative probability of the occurrence of mutation $i$ in gene $j$, we define the UNEECON score of mutation $i$ as $1 - \mathbb{E}_{u_j | \text{Data}}(\eta_{ij} | u_j)$. The UNEECON-G score is defined as the average UNEECON score of all the missense mutations in a gene.

## Distributions of UNEECON scores across gene categories and protein regions

We investigated the distributions of UNEECON scores across different gene categories and protein regions. We obtained lists of haploinsufficient genes [37] ($n = 294$), autosomal dominant disease genes [35, 36] ($n = 709$), autosomal recessive disease genes [35, 36] ($n = 1,183$), and olfactory receptor genes [45] ($n = 371$) from the GitHub repository for the MacArthur Lab at the Broad Institute (https://github.com/macarthur-lab/gene_lists). We obtained annotations of $\alpha$-helices, $\beta$-strands, hydrogen-bonded turns, enzyme active sites, and binding sites from UniProt [47]. We obtained annotated disordered protein regions from MobiDB 3.0 [48]. We plotted the distributions of UNEECON scores using R [75].

## Comparison with other methods in the prediction of ClinVar pathogenic variants

We evaluated the performance of UNEECON, compared with eight previously published methods, in the setting of predicting pathogenic missense variants in the ClinVar database downloaded on Feb 25, 2019 [30]. We obtained MPC [25], PrimateAI [11], Eigen [50], CADD [8], RVIS [16], and pLI [17] scores from dbNSFP (version 4.0b2a) [79]. We obtained LASSIE scores [12] from the LASSIE GitHub repository (https://github.com/CshlSiepelLab/LASSIE) and CCR score (version 2) [22] from https://s3.us-east-2.amazonaws.com/ccrs/ccrs/ccrs. autosomes.v2.20180420.bed.gz. It is worth noting that we used the version of pLI scores trained on the gnomAD exome sequencing data [29].

We considered autosomal missense variants with annotations of "pathogenic/likely patho-genic" as positives and the ones with annotations of "benign/likely benign" as negative controls. We further removed the variants with conflict pathogenicity annotations and the variants present in the gnomAD exome sequencing dataset [29]. For all the comparisons, we only included the variants scored by all methods. We plotted the receiver operating characteristic (ROC) curves and calculated the areas under the receiver operating characteristic curves (AUCs) using the ROCR package [80]. Because the AUC metric is sensitive to label imbalance, we matched the numbers of positives and negatives using random sampling without replacement. After all the filtering steps, we obtained 473 pathogenic and 473 benign variants in the autosomal dominant dataset, as well as 277 pathogenic and 277 benign variants in the autosomal recessive dataset.

## Comparison with alternative methods in predicting *de novo* mutations associated with developmental disorders

We downloaded *de novo* mutations identified in 4,293 individuals affected by developmental disorders [31] and 2,278 healthy individuals from denovo-db (version v1.6.1). The healthy controls included individuals enrolled in previous studies of autism [81–85], severe non-syndromic sporadic intellectual disability [86], schizophrenia [87], and healthy populations [88–91]. We removed redundant variants and any variants presented in the gnomAD exome sequencing data. Then, for various percentile rank cutoffs (top 10%, 20%, 30%, and 40%), we calculated the $\log_2$ odds ratio of the enrichment of predicted deleterious variants in affected individuals using the fisher.test function in R [75].

## Comparison with alternative methods in predicting disease genes and essential genes

We obtained four sets of disease genes and essential genes from the GitHub repository for the MacArthur Lab at the Broad Institute (https://github.com/macarthur-lab/gene_lists). These gene sets included 294 haploinsufficient genes [37], 709 autosomal dominant disease genes [35, 36], 2,454 human orthologs of mouse essential genes [33, 34], and 683 essential genes based on CRISPR screen data from human cell lines [32]. For each set of the haploinsufficient, autosomal dominant, and mouse essential genes, we utilized the other autosomal genes as negative controls. Because it is easier to reject the null model of neutral evolution in a longer gene, gene constraint scores tend to be correlated with gene length [17, 29]. To control for the impact of gene length on performance evaluation, we used MatchIt [92] to pair each disease gene with a non-disease gene with matched gene length, resulting in a negative gene set with matched gene number and gene length. Similarly, for the 683 human essential genes based on CRISPR screen data, we used the 913 nonessential genes from the same study [32] as negative controls and constructed a negative gene set with matched gene length and gene number. We plotted the ROC curves and calculated the AUC metrics using the ROCR package [80].

## Comparison of UNEECON-G and pLI scores

To evaluate the relationship between gene-level intolerance to missense mutations and that to loss-of-function mutations, we compared the UNEECON-G scores with the pLI scores trained on the gnomAD data [17, 79]. We identified 1,912 genes intolerant to loss-of-function mutations based on a threshold of pLI score of 0.9. Then, we split the 1,912 genes into 956 genes tolerant to missense genes and 956 genes intolerant to missense mutations based on the median UNEECON-G score. We used PANTHER [93] to calculate the enrichment of Reactome

pathways and Gene Ontology categories in the set of genes intolerant to both missense and loss-of-function mutations, compared with those tolerant to missense mutations but not to loss-of-function mutations. We also downloaded 1,054 autism related genes from the SFARI database on March 9, 2019 [59], and evaluated the enrichment of autism genes in the two gene sets.

## Supporting information

**S1 File. Supplementary material.** Supplementary figures and tables.
(PDF)

**S2 File. Supplementary dataset.** List of 1,912 genes intolerant to loss-of-function mutations.
(TSV)

## Acknowledgments

The author thanks Noah Dukler for comments on the manuscript.

## Author Contributions

**Conceptualization:** Yi-Fei Huang.

**Data curation:** Yi-Fei Huang.

**Formal analysis:** Yi-Fei Huang.

**Funding acquisition:** Yi-Fei Huang.

**Investigation:** Yi-Fei Huang.

**Methodology:** Yi-Fei Huang.

**Project administration:** Yi-Fei Huang.

**Resources:** Yi-Fei Huang.

**Software:** Yi-Fei Huang.

**Validation:** Yi-Fei Huang.

**Visualization:** Yi-Fei Huang.

**Writing – original draft:** Yi-Fei Huang.

**Writing – review & editing:** Yi-Fei Huang.

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
