## [Decision Letter · Decision Letter 0]

10 Jan 2020

Dear Dr Huang,

Thank you very much for submitting your Research Article entitled 'Unified inference of missense variant effects and gene constraints in the human genome' to PLOS Genetics. Your manuscript was fully evaluated at the editorial level and by independent peer reviewers. The reviewers appreciated the attention to an important problem, but raised some substantial concerns about the current manuscript. Based on the reviews, we will not be able to accept this version of the manuscript, but we would be willing to review again a much-revised version. We cannot, of course, promise publication at that time.

If you decide to revise the manuscript for further consideration at PLOS Genetics, please aim to resubmit within the next 60 days, unless it will take extra time to address the concerns of the reviewers, in which case we would appreciate an expected resubmission date by email to plosgenetics@plos.org.

[LINK]

We are sorry that we cannot be more positive about your manuscript at this stage. Please do not hesitate to contact us if you have any concerns or questions.

Yours sincerely,

Scott M. Williams

Section Editor: Natural Variation

PLOS Genetics

Hua Tang

Section Editor: Natural Variation

PLOS Genetics

Reviewer's Responses to Questions

**Comments to the Authors:**

Reviewer #1: This paper describes the development and evaluation of UNEECON, a framework for jointly predicting deleterious variants and constrained genes. This is certainly an interesting topic in the context of variant effect prediction and interpretation. I find the attempt to unify variant-level and gene-level quite innovative, and it is certainly an approach that will be useful in the study of severe, early onset disorders. Further, the use of a deep neural network to learn parameters relevant to population genetics from millions of variants from gnomAD is a novel contribution to this area. From the perspective of constrained gene prediction, UNEECON results are quite promising. However, I remain skeptical of some of the claims with regards to pathogenicity prediction and the overall argument that this method would be better in practice than those evaluated here (see below). Overall, the paper is clearly written and the methods are outlined satisfactorily (see minor comments for some things that need to be clarified). I outline my comments below:

MAJOR:

- A general issue that has plagued the field is the problem of unevenly distributed variant information across genes. Some genes are over-studied and are likely to have more variants identified as pathogenic. More importantly, many genes are likely to contain variants from only one class. UNEECON is interesting in this context that it is trained on genes that are mostly going to contain only benign variants and is evaluated on a ClinVar set that will skew mostly towards pathogenic-only or bi-class genes. Ref. 58 from this paper highlighted a method that performed extremely well in its evaluations when run on “pathogenic-only” or “benign-only” proteins but drastically underperformed on “mixed” genes. Given that UNEECON is heavily influenced by gene-level features, I wonder if it is susceptible to the same issue. One way to test this would be to perform a version of the ClinVar evaluation on only the subset of genes that contain both classes of variants. If performance drops then perhaps unification of variant-level and gene-level information may not be the best approach for variant pathogenicity prediction.

- On a related note, I am concerned about information leakage between the training set and evaluation sets used in this paper. I agree that UNEECON benefits from actually not using the pathogenic variants in its training and evaluation. However, the sheer size of gnomAD is expected to include every known gene in the training of the deep neural network. Since UNEECON uses gene-level information, there is a distinct possibility that performance may be inflated even after excluding variants in both ClinVar and gnomAD. In fact, Ref. 58 (cited for the circularity and inflation issues) points this out and recommends gene-level partitioning for cross-validation experiments such as those conducted by PolyPhen-2 and MutPred. Of course, in the context of this paper, the proposed experiment would be to train a version of UNEECON without variants in genes from the ClinVar set and evaluate that version on the ClinVar set. That way any gene-level bias in the performance measures would be eliminated.

- On page 8, line 254, is it all that surprising that UNEECON-G and pLI scores do not correlate? Intuitively, the impact of missense variants and LOF mutations are going to vary in magnitude even within the same gene. While a biological explanation (as provided here) for this may very well be plausible, it is more likely that the discrepancies are due to technical reasons. A recent commentary (PMID: 30977936) has touched upon issues related to the methodology and applicability of pLI scores. This commentary highlights the example of BRCA genes that have near-zero pLI scores but are known to harbor several deleterious missense variants.

MINOR:

- The bimodality of the UNEECON score distribution for active sites is worrisome with the peak closer to 0.25 is a little confusing. I interpret this as “there are more variants in active sites that have low UNEECON scores than high.” This is counter-intuitive and warrants some explanation.

- In the functional analyses related to Fig. 5, are there any interesting depletions? I am curious about the functions of those genes that are tolerant to missense but not to LOF mutations. I am also not sure what “unclassified” means in this context.

- What is the difference between Eqns. 2 and 3? It is difficult to tell with q_i being defined.

- In the Methods section, it would be helpful to readers if a clear account of the parameters to be estimated is provided up front.

- The paper is missing details of the final model that emerged from the evaluation process, its architecture and its parameters.

- Similarly, the paper lacks details on dataset sizes, particularly in the context of model training and evaluation. How many variants were used to train the deep mixed-effects model? How many variants were included in the evaluations relevant to ClinVar? How many pathogenic and how many benign?

- I am also curious about the activation function of the output layer of the neural network. This is of particular relevance to z_ij and its scaling relative to u_j. Is there a potential for one quantity to systematically dominate the other in Eqn. 9?

Reviewer #2: The work represents an important advance in prioritization of genes and variants relevant to human disease. it has been known since the introduction of gene level intolerance scoring in 2013 that gene level metrics of the strength of purifying selection provide independent information about variation pathogenicity to the longer established variant level metrics that largely depend on conservation and amino acid substitution features. While attempts have been made previously to integrate both approaches into a single predictive framework these have been based on supervised learning approaches using a set of putatively pathogenic and benign variants. The work here combines a selected set of variant level features with a gene level term and estimates selective constraint operating against all possible gene sequence changes based on human polymorphism data compared against sequence specific mutability. As such, it provides an integrated approach assessing purifying selection operating in the human population.

The authors have rerun the standard assessments used to test both gene level and variant level predictors with generally improved performance both for identifying relevant gene sets (e.g. haploinsufficient genes) and pathogenic variants. In addition to these advances, the model allows some novel biological insights, including explaining an important reasons for discrepancy between intolerance to missense and loss of function variation as being due to the proportion of proteins that is disordered. The model also highlights that the gene level term is more informative than variant level terms which is still not as widely appreciated as it should be.

For these reasons the work here represents an important advance in the field.

While the paper is generally clearly written and the conclusions generally fair, I do have a couple of relatively minor suggestions for consideration. Perhaps most fundamentally, while the use of UNEECON deep learning model to combine variant features and a gene level term to predict the strength of selection operating against specific alleles is welcome, since it allows non linear combinations of these terms to be learned, it is striking that a linear approximation of the UNEECON model is very highly correlated, suggesting little benefit from the model learning optimum non linear combinations. The authors appropriately use the linear model to infer the relative importance of features, but the very high correlation between the two models suggests the linear modle is likely to have similar performance to UNEECON. Given the more direct interpretability of the linear model, the authors should comment on whether the more complex model is in fact needed for use. The second small point is that some of the comparisons are inappropriate since some of the metrics are used in ways they were not intended for. For example, in Figure 3a representing prediction in distinguishing pathogenic variants gene level metrics such as RVIS are compared directly to UNEECON. As outlined however in the initial work, gene level metrics are intended to be used alongside some version of a variant level predictor (since as emphasized here and in the original publications the two approaches offer independent information). The fair comparison therefore for generating a version of figures 3 focused on variants would be to use a combination of a variant and gene level metric for all those comparisons like RVIS that are gene level metrics. This idea was outlined in the initial publications under the banner of a combined threshold for both gene level and variant level. I have no doubt that UNEECON would still perform better, but one appropriate simple comparison would be to re run these analyses including for example a hard threshold on some appropriate variant score such as PP2 alongside the quantitative gene level score such as RVIS as currently used. Finally, the gene level metrics in use are known to struggle with small genes since there is often not enough polymorphism data to infer selection. The authors should address robustness to gene size.

**Have all data underlying the figures and results presented in the manuscript been provided?**

Reviewer #1: Yes

Reviewer #2: Yes

PLOS authors have the option to publish the peer review history of their article (what does this mean?). If published, this will include your full peer review and any attached files.

Reviewer #1: No

Reviewer #2: No

---

## [Decision Letter · Decision Letter 1]

30 Apr 2020

Dear Dr Huang,

Thank you very much for submitting your Research Article entitled 'Unified inference of missense variant effects and gene constraints in the human genome' to PLOS Genetics. Your manuscript was fully evaluated at the editorial level and by independent peer reviewers. The reviewers appreciated the attention to an important topic but identified some aspects of the manuscript that should be improved.

We therefore ask you to modify the manuscript according to the review recommendations before we can consider your manuscript for acceptance. Your revisions should address the specific points made by each reviewer.

[LINK]

Yours sincerely,

Scott M. Williams

Section Editor: Natural Variation

PLOS Genetics

Hua Tang

Section Editor: Natural Variation

PLOS Genetics

Reviewer's Responses to Questions

**Comments to the Authors:**

Reviewer #1: The revised version of the paper addresses most of the concerns that I had with the original version of the paper. However, I still remain skeptical of the claim of UNEECON’s “unmatched” performance when it comes to pathogenicity prediction. Although the AUCs are indeed higher for UNEECON in Figs. 3, 4, S3 and S4, performances in the most important region of the ROC curves (the low-false-positive-rate region) tend to be on comparable to other methods. I suggest toning down such strong claims made with regards to performance of UNEECON in pathogenicity prediction, when compared to other methods.

I also would like to follow up on the following statement in the item-by-item response: “Training a version of UNEECON without gnomAD variants in disease genes will disable UNEECON’s ability to learn gene-level constraints in disease genes, leading to an underestimation of UNEECON’s performance.” This gets to the actual motivation behind my comment. If gene-level constraints are that important to UNEECON’s performance, then it is expected that UNEECON will underperform when attempting to predict a pathogenic variant in a gene with no previous known disease association. The gnomAD subset that does not overlap with ClinVar serves as a proxy for such genes as it is quite comprehensive in the coverage of the genome. My original concern was that UNEECON may simply be good at separating disease-associated genes (which is as the author correctly said is subject to ascertainment bias) from those in gnomAD, and that this was a major driver of variant-level predictive performance. This is somewhat alleviated through the inclusion of Fig. S4 but a true test of UNEECON’s ability to contribute to novel discoveries is in its ability to make correct variant-level predictions in “undiscovered” disease genes. If an experiment to test this seems infeasible, it would be helpful to clearly state this as a limitation of the model in the Discussion section.

Reviewer #2: the authors have done a thorough job of responding to the reviews and I have no further comments

**Have all data underlying the figures and results presented in the manuscript been provided?**

Reviewer #1: Yes

Reviewer #2: Yes

PLOS authors have the option to publish the peer review history of their article (what does this mean?). If published, this will include your full peer review and any attached files.

Reviewer #1: No

Reviewer #2: No

---

## [Editor Report · Decision Letter 2]

9 Jun 2020

Dear Dr Huang,

We are pleased to inform you that your manuscript entitled "Unified inference of missense variant effects and gene constraints in the human genome" has been editorially accepted for publication in PLOS Genetics. Congratulations!

Yours sincerely,

Scott M. Williams

Section Editor: Natural Variation

PLOS Genetics

Hua Tang

Section Editor: Natural Variation

PLOS Genetics

Comments from the reviewers (if applicable):

**Data Deposition**

http://datadryad.org/submit?journalID=pgenetics&manu=PGENETICS-D-19-01659R2

**Press Queries**

---

## [Editor Report · Acceptance letter]

7 Jul 2020

PGENETICS-D-19-01659R2 

Unified inference of missense variant effects and gene constraints in the human genome 

Dear Dr Huang, 

We are pleased to inform you that your manuscript entitled "Unified inference of missense variant effects and gene constraints in the human genome" has been formally accepted for publication in PLOS Genetics! Your manuscript is now with our production department and you will be notified of the publication date in due course.

With kind regards,

Matt Lyles

PLOS Genetics

On behalf of:
